# Osr1 Is Required for Mesenchymal Derivatives That Produce Collagen in the Bladder

**DOI:** 10.3390/ijms222212387

**Published:** 2021-11-17

**Authors:** Vasikar Murugapoopathy, Philippe G. Cammisotto, Abubakr H. Mossa, Lysanne Campeau, Indra R. Gupta

**Affiliations:** 1Department of Human Genetics, McGill University, Montreal, QC H3A 0C7, Canada; vasikar.murugapoopathy@mail.mcgill.ca; 2Lady Davis Research Institute for Medical Research, Jewish General Hospital, Montreal, QC H3T 1E2, Canada; philippe.cammisotto.1@ulaval.ca (P.G.C.); abubakr.mossa@mail.mcgill.ca (A.H.M.); lysanne.campeau@mcgill.ca (L.C.); 3Division of Urology, Department of Surgery, Jewish General Hospital, McGill University, Montreal, QCH3T 1E2, Canada; 4Research Institute of the McGill University Health Center, Montreal, QC H3H 2R9, Canada; 5Department of Pediatrics, McGill University, Montreal, QC H4A 3J1, Canada

**Keywords:** Odd1, extracellular matrix, bladder disease, bladder development

## Abstract

The extracellular matrix of the bladder consists mostly of type I and III collagen, which are required during loading. During bladder injury, there is an accumulation of collagen that impairs bladder function. Little is known about the genes that regulate production of collagens in the bladder. We demonstrate that the transcription factor Odd-skipped related 1 (Osr1) is expressed in the bladder mesenchyme and epithelium at the onset of development. As development proceeds, Osr1 is mainly expressed in mesenchymal progenitors and their derivatives. We hypothesized that Osr1 regulates mesenchymal cell differentiation and production of collagens in the bladder. To test this hypothesis, we examined newborn and adult mice heterozygous for *Osr1*, *Osr1^+/−^*. The bladders of newborn *Osr1^+/−^* mice had a decrease in collagen I by western blot analysis and a global decrease in collagens using Sirius red staining. There was also a decrease in the cellularity of the lamina propria, where most collagen is synthesized. This was not due to decreased proliferation or increased apoptosis in this cell population. Surprisingly, the bladders of adult *Osr1^+/−^* mice had an increase in collagen that was associated with abnormal bladder function; they also had a decrease in bladder capacity and voided more frequently. The results suggest that Osr1 is important for the differentiation of mesenchymal cells that give rise to collagen-producing cells.

## 1. Introduction

The bladder is a hollow sac that stores urine and can expand up to three to four times its size when full. When the bladder muscle contracts, the bladder collapses and empties. The contractions are coordinated by three primary layers: the epithelial layer, the lamina propria, and the muscle layer. The latter two layers are rich in extracellular matrix. The extracellular matrix (ECM) bears most of the mechanical load of the bladder, which is important to maintain low pressure [1,2,3,4,5]. When bladder drainage is obstructed either anatomically or functionally, as seen in spinal cord injuries, muscle contractions continue but they are no longer synchronized with urethral sphincter opening. The bladder wall becomes thicker with a marked increase in ECM in the lamina propria and muscle [6,7,8,9]. This results in a high-pressure bladder that is stiff and unable to empty. Patients with this bladder dysfunction require intermittent catheterization to maintain urinary continence.

To understand ECM deposition in the bladder, the molecular and cellular origins of bladder development need to be examined. The patterning of the bladder relies on epithelial to mesenchymal cross talk between the endoderm-derived epithelia and the mesoderm-derived tail bud mesenchyme [10]. Sonic hedgehog (Shh) is secreted by epithelial cells and diffuses across the bladder to specify the adjacent mesenchyme [11]. The higher concentration of Shh adjacent to the epithelium results in differentiation of cells to form the lamina propria, while the lower concentration results in differentiation of cells into the muscle layer. Most ECM is laid down postnatally in the bladder, which correlates temporally with increased urine volume and the need for greater bladder capacity. While much is known about the embryonic formation of the muscle and lamina propria layers [11,12,13,14,15], it is unclear what regulates postnatal deposition of ECM.

Odd-skipped related 1 (Osr1) is a zinc-finger transcription factor that regulates mesenchymal cell survival and differentiation in the lungs, heart, tongue, limbs and kidneys during embryonic development [16,17,18,19,20,21]. Osr1^−/−^ mice die around E12 and show severe defects in mesenchymal cell patterning. Osr1 has been shown to function directly downstream of Gli, the primary effector of Shh signaling [22]. In the limbs and liver, Osr1 is required for deposition of ECM that serves as a niche for mesenchymal cells. In both of these tissues, Osr1 is expressed in fibrogenic progenitor cells that become activated during injury [23,24].

Previously, we showed that Osr1 is expressed in the mesenchyme and epithelium of the primordial bladder at E10.5, and that its expression continues in the developing bladder until birth [25]. Based on the expression pattern of Osr1 and its interaction with key signaling pathways involved in bladder development, we hypothesized that Osr1 is a critical regulator of extracellular matrix production and mesenchymal cell differentiation in the bladder. Here we show a detailed analysis of collagen expression in *Osr1^+/−^* mice from the embryo to the adult, and we examine bladder function in adult mice.

## 2. Results

### 2.1. Osr1 Expression Changes during Bladder Development

Our previous studies showed strong expression of Osr1 in the developing bladder using an *Osr1* reporter mouse line in which the *Osr1* null allele is fused to a β-galactosidase reporter [25]. Due to the lack of a good antibody, we used fluorescent in situ hybridization (RNAScope) to track endogenous *Osr1* mRNA expression throughout development (Figure 1). An antisense probe for *vimentin* (*vim*) was used to label mesenchymal cells in the embryonic bladder and fibroblasts in newborns and adults. Another probe for *alpha-smooth muscle actin* (*acta2*) was used to label smooth muscle cells. The bladder layers were defined by *vim* and *acta2* expression: the epithelial layer was *vim−*, *acta2−*, while the lamina propria was *vim+*, *acta2−*, and the smooth muscle layer was *vim+*, *acta2*+. At embryonic (E) day 14, when the bladder mesenchyme differentiates to form the lamina propria and muscle layer, *Osr1* was strongly expressed in all cells in the epithelial and smooth muscle layers, with a few cells expressing *Osr1* in the lamina propria layer. Postnatally, there is an increase in expression of extracellular matrix proteins, which correlates with increased urine volume and the need for greater bladder distensibility [26]. At postnatal (P) day 1, *Osr1* expression was detected in almost all cells in the lamina propria and epithelia, and some cells in the muscle layer. At 5 weeks of age, bladder function is mature. Mice have acquired the ability to void under voluntary control, and no longer rely on the mother to stimulate involuntary bladder contractions. At this timepoint, Osr1 was predominantly expressed in fibroblasts in the lamina propria and in smooth muscle cells and presumptive myofibroblast cells in the muscle layer. The expression analysis demonstrates that Osr1 is predominantly expressed in derivatives of the bladder mesenchyme, which give rise to ECM-producing cells postnatally.

### 2.2. Osr1^+/−^ Newborn Mice Exhibit Decreased Cellularity in the Lamina Propria Layer

To study the function of Osr1 in bladder mesenchyme, we examined mice bearing a heterozygous loss-of-function allele in *Osr1*, because null mice die prior to the formation of the bladder [19]. To determine if Osr1 regulates mesenchymal differentiation, immunofluorescent detection of Vimentin and α-SMA protein was performed in newborn *Osr1^+/−^* and *Osr1^+/+^* mice. There was a reduction in Vimentin-expressing cells in the lamina propria of *Osr1^+/−^* mouse bladders, suggesting there could be a depletion in fibroblast populations (Figure 2). Vimentin-positive cells were quantified per unit area in the lamina propria, and this demonstrated a significant decrease in cell number in newborn *Osr1^+/−^* mice compared to their *Osr1^+/+^* littermates (expressed in cells/μm^2^ as mean +/− standard deviation 0.008 +/− 0.0019 vs. 0.005 +/− 0.0016, *p* = 0.038). Immunofluorescent detection of α-SMA protein and western blot analysis revealed no differences between *Osr1^+/−^* and *Osr1^+/+^* mice (Appendix A).

When *Osr1* is removed in mesenchymal cells in other embryonic tissues such as the embryonic kidney, there is impaired cell survival [17]. To determine if loss of *Osr1* resulted in changes in mesenchymal cell survival or proliferation in the bladder, we performed TUNEL and Ki67 staining, respectively, at E16 and P1 in the mouse bladder. At both timepoints, both *Osr1^+/−^* and *Osr1^+/+^* mice showed little to no cell death in the bladder (Appendix A). The amount of cell proliferation in the bladder was also similar at both timepoints when comparing *Osr1^+/−^* and *Osr1^+/+^* embryos and newborn pups (Figure 3). These results suggest that the loss of cellularity in the lamina propria layer could be due to a defect in mesenchymal cell differentiation.

### 2.3. Osr1^+/−^ Newborn Mice Exhibit Decreased Collagen in the Bladder

Collagen I and III are secreted by fibroblasts and are the most abundant ECM proteins in the bladder [1]. Most of the ECM in the lamina propria is laid down postnatally. To determine if the decreased cellularity in the lamina propria resulted in a decrease in collagen proteins, we performed Sirius red staining in newborn *Osr1^+/−^* and *Osr1^+/+^* mice. Using Sirius red staining imaged under birefringent light, there was a decrease in thick (typically collagen I) and to a lesser extent, thin (predominantly collagen III) fibrils in the lamina propria and muscle layers in *Osr1^+/−^* mice (Figure 4A,B). To quantify the amount of collagen I and III protein, western blots of whole bladder lysates were performed using glyceradehyde-3-phosphate dehydrogenase (GAPDH) as a loading control (Figure 4C–F). There was significantly less collagen I protein in *Osr1^+/−^* mice compared to *Osr1^+/+^* littermates (expressed using densitometry as relative density, mean +/− standard deviation Osr1^+/+^: 1.18 +/− 0.59 vs. Osr1^+/−^: 0.508 +/− 0.35, *p* = 0.05). To determine if Osr1 affected transcription of collagen genes, we performed qRT-PCR for *col1a1* (collagen I) and *col3a1* (collagen III) mRNA in bladders of *Osr1^+/−^* and *Osr1^+/+^* mice, using *gapdh* as a control. There was no significant difference in transcript levels of either *collagen I* or *collagen III* (Appendix A), suggesting that the differences seen in the western blot are primarily due to the decreased number of fibroblasts. Taken together, the decrease in Vimentin-positive cells in the lamina propria correlates with a decrease in collagen proteins throughout the bladder wall.

### 2.4. Osr1^+/−^ Adult Mice Exhibit Increased Collagen in the Bladder

To determine if the decrease in collagen at the newborn stage affected the composition of the adult bladder, Masson’s trichrome and Sirius red staining for collagen was performed (Figure 5A–D). Surprisingly, there was an increase in collagen, especially in the muscle layer of *Osr1^+/−^* mice. Indeed, when quantifying collagen from adult bladder lysates using western blot analysis (Figure 5E–H), there was no longer a significant decrease in collagen I, suggesting a relative increase in collagen deposition between the newborn and adult period in *Osr1^+/−^* mice. Collagen III levels were comparable by western analysis. We then analyzed cellularity of the lamina propria using Vimentin immunofluorescence, as done in the newborns (Figure 6). In the adults there were similar cell density in the lamina propria of both Osr1+/+ and Osr1 +/− mice (expressed in cells/μm^2^ as mean +/− standard deviation 0.0035 +/− 0.0004 vs. 0.0039 +/− 0.0001, *p* = 0.16). We also noted a subtle increase in Vimentin-positive cells in the muscle layer of Osr1^+/−^ compared to Osr1^+/+^ bladders (expressed as a fraction of Vimentin-positive cells / DAPI-positive cells, as mean +/− standard deviation 0.23 +/− 0.03 vs. 0.18 +/− 0.01, *p* = 0.09). We speculate that *Osr1^+/−^* mice may compensate for the decrease in collagen in the newborn period with overdistension of the bladder which, in turn, may contribute to the expansion of fibroblast populations and the increased deposition of collagen within the muscle layer in adult Osr1^+/−^ mice.

### 2.5. Osr1^+/−^ Adult Mice Have Poor Bladder Function

To determine if the increased collagen in the bladders of adult *Osr1^+/−^* mice results in abnormal function, cystometry was performed in *Osr1^+/−^* and *Osr1^+/+^* mice by surgical insertion of a suprapubic catheter (Table 1). *Osr1^+/−^* mice had significantly decreased micturition volumes, intercontraction intervals, bladder capacities and bladder compliances when compared to their wild-type littermates (*p* < 0.05). Both male and female *Osr1^+/−^* mice exhibited these phenotypes when compared to sex-matched *Osr1^+/+^* mice (Appendix A). Taken together, the increase in collagen in the bladders of adult *Osr1^+/−^* mice results in a stiffer, smaller bladder that results in increased urinary frequency. 

## 3. Discussion

We show that the heterozygous loss of Osr1 results in abnormal extracellular matrix deposition in the bladder that is associated with impaired function. While previous studies have shown that Osr1 is important for embryonic development, we show that it is also expressed in the bladder during adulthood. Osr1 is expressed in epithelial and mesenchymal cell lineages during embryonic bladder development, but postnatally it is expressed predominantly in mesenchymal cell derivatives. The heterozygous loss of Osr1 resulted in decreased cellularity in the lamina propria layer, which originates from mesenchymal cells. The decrease in cellularity was accompanied by a decrease in collagen. Surprisingly, an increase in collagen deposition was observed in the bladders of adult *Osr1^+/−^* mice which also had bladder dysfunction. In summary, Osr1 regulates a mesenchymal cell population that is important for ECM deposition in the bladder in the embryo and in the adult.

Osr1 regulates the differentiation of mesenchymal progenitor populations into fibro-adipogenic cells in the limb and peribiliary fibroblasts in the liver [20,23]. In the embryonic limb, loss of Osr1 resulted in a depletion of fibro-adipogenic cells due to a decrease in proliferation and cell survival [20]. Our data shows that Osr1 is expressed in embryonic mesenchymal progenitor cells that later become fibroblasts in the mature bladder. Although we did not see differences in cell death or proliferation at E16 or P1, we cannot completely exclude the possibility that there was a depletion of mesenchymal cells between these two timepoints. Osr1 transcriptionally regulates the production of collagens, proteoglycans, and other ECM components [20], as *Osr1* null mice had lower mRNA expression of these genes by qRT-PCR. We, however, did not see differences in mRNA levels of *col1a1* (*or col3a1*) at the newborn period, despite the differences at the protein level between heterozygous and wildtype mice, which suggests that Osr1 is not regulating transcription of collagens in the bladder. It is important to note that the limb model was a conditional null, while ours is a haploinsufficiency in the bladder. Taken together, the decrease in collagen observed in the newborn bladder results from the depletion of the fibroblast lineage. We speculate that in the bladder, Osr1 may be required for the formation of a stem cell niche that directs the differentiation of fibroblasts as observed in the limbs and liver.

The bladder mesenchyme gives rise to several cell types, including smooth muscle cells, neurons, immune cells, and several fibroblast populations [27]. A single-cell transcriptomic analysis of the human and mouse bladder revealed Osr1 expression in several different fibroblast and interstitial cell populations in the adult bladder [27]. The fibroblasts located directly underneath the basal epithelium, also called the suburothelial fibroblasts, are critical for mediating epithelial to mesenchymal communication during bladder development [28]. Postnatally, these cells propagate contraction signals to the muscle to initiate voiding. Osr1^+/−^ mice have a decrease in the number of cells within the lamina propria layer, including the suburothelial cells, which may affect both bladder development and voiding function. The defects in the adult Osr1^+/−^ mouse bladder may not only be due to the initial lack of collagen but also due to the loss of signaling from suburothelial cells.

We used a number of methods to detect collagen in the newborn and adult bladder, because they each provide different information. Masson’s trichrome, a common stain for collagen, was useful in detecting the abundant collagen seen in adult mice, however, it was not sensitive enough to visualize the lower amount of collagen seen at the newborn stage. Sirius red staining when imaged under a polarized light filter can distinguish thick and thin collagen fibrils which usually represent collagen I and collagen III fibrils, respectively. Sirius red was more sensitive and could be used to detect collagen in the newborn bladder. Quantification of Sirius red staining is difficult in the bladder, as levels of green, red, and yellow seen under polarized light can vary depending on the angle of the polarized light and the orientation of the fibrils [29]. As the lamina propria consists of collagens interwoven in a complex mesh structure, it is difficult to standardize images for quantification. To quantify collagen, we used western blot analysis. One limitation of this method, however, is that differences in specific bladder layers cannot be appreciated. Indeed, in the bladders of adult *Osr1^+/−^* mice, we saw an increase in collagen in the smooth muscle layer using Masson’s trichrome staining and Sirius red staining, however total bladder content of Collagens I and III were similar by western analysis when compared to *Osr1^+/+^* mice.

It is intriguing as to why the newborn Osr1^+/−^ mice have a decrease in collagen in the lamina propria layer, while the adult Osr1^+/−^ mice have an increase in collagen in the muscle layer. Due to their small size and inability to undergo surgery, we were unable to perform cystometry on newborn mice, but in adult mice, the increase in collagen observed in the bladders of Osr1^+/−^ mice correlated with bladder dysfunction. Adult Osr1^+/−^ mice had smaller bladder capacities and urinated much more frequently. We speculate that this increase in collagen could be part of an injury response in the adult bladder. Collagens I and III in the lamina propria of the bladder are critical for bearing the mechanical load during filling. Due to the lack of collagen in the lamina propria, Osr1^+/−^ pups likely have a weak atonic bladder that is unable to empty [2].This would put a mechanical burden on the smooth muscle layer, similar to what is seen with functional bladder obstruction. Bladder obstruction has been shown to result in dysregulation of fibroblasts, and loss of ECM modulatory proteins such as MMPs, which results in collagen accumulation [30]. Once collagen accumulates, bladder function is perturbed, resulting in decreased bladder capacity and increased voiding frequency, as seen in adult Osr1^+/−^ mice. Interestingly, Osr1 has been shown to be expressed in populations of fibroblasts during injury in the limb [24]. Our results cannot distinguish if Osr1 is required for fibroblast maintenance or activation during injury. This type of question would be best addressed by performing lineage tracing of Osr1-positive cells during development and during injury. Therefore, elucidating the role of Osr1 in the bladder may have therapeutic implications in understanding how fibroblasts function during fibrosis.

## 4. Methods

### 4.1. Mouse Lines

*Osr1^tm1Jian^* or *Odd1-LacZ* were obtained from Jackson Laboratories and maintained on a C57BL/6J background (Jackson Lab #009387). These mice contain a β-galactosidase reporter fused to an *Osr1* null allele. These mice were compared to Osr1^+/+^ littermates in all analyses. All mice were housed in ventilated cages with up to five mice/cage with wood chip bedding and a diurnal light cycle providing 12 h of light. The mice fed and drank from their water bottles ad libitum with feed provided as per the US National Research Council recommendations for rodent nutrition. Environmental enrichment with either paper strands or cellulose-based shelters were included in each cage. The room temperature was maintained between 18 and 24 °C, and the humidity was maintained between 30 and 70%. All animal studies were performed in accordance with the regulations of the Canadian Council on Animal Care and approved by the McGill University Animal Care Committee (AUP 4120).

### 4.2. Tissue Collection and Processing

Mice were euthanized at E14, E16, P1 and five weeks of age, and bladders were dissected into cold PBS and fixed in 4% PFA overnight for sectioning or snap-frozen in liquid nitrogen for western blots or qRT-PCR. Fixed tissues were then transferred to 70% ethanol for long term storage at 4 °C. Some tissues were processed for paraffin embedding as previously described [31]. Bladders were sectioned at 5 um thickness.

### 4.3. RNAScope for mRNA Expression

Using the multiplex fluorescent V2 kit and the 4 plex ancillary kit [RNAScope, ACD Biosystems, Newark, CA, USA] mRNA detection was performed for *acta2* for alpha-smooth muscle actin, *vim* for vimentin, and *osr1* for Odd-skipped related 1. Staining was performed according to manufacturer’s directions.

### 4.4. Sirius Red Staining

Paraffin sections of mouse bladders were deparaffinized and rehydrated in water before staining with Sirius red reagent (Polysciences Inc, Warrington, PA, USA). Sections were then washed in acetic acid solution and dehydrated in ethanol and xylene before cover slipping. Sections were imaged under brightfield light to show total collagen (red) and birefringent light to discern thick (yellow) and thin (green) collagen fibrils. Images were taken using the Zeiss Axiovert3 microscope.

### 4.5. Immunofluorescent Staining

Paraffin sections were deparaffinized and rehydrated in PBS. Heat-mediated citrate antigen retrieval was performed, and sections were blocked in 1% BSA for 1 h before proceeding to staining. Primary antibody incubation was performed at 4 °C overnight for rabbit anti-mouse alpha-smooth muscle actin (Abcam, Toronto, ON, Canada), rabbit anti- Ki67 (Abcam) and mouse anti-mouse vimentin (Santa Cruz, Dallas, TX, USA) at a dilution of 1/100. Sections were then washed and incubated with secondary antibodies (Invitrogen, Waltham, MA, USA) as follows, anti-rabbit 488, anti-rabbit 555, or anti-mouse 555 for 1 h at room temperature at 1/500 dilution. Sections were then washed and mounted with antifade reagent (ProLong Gold Antifade Mountant). Sections were imaged under confocal microscopy (Leica LSM880, Richmond Hill, ON, Canada). Image processing was done using the Zen black software and Fiji.

### 4.6. TUNEL

The TUNEL assay was performed on paraffin sections. Samples were deparaffinized and re-hydrated to PBS. The Roche In Situ Cell Death Detection Kit, TMR red was used and protocol was performed as listed on datasheet (version 12) (Roche, Laval, QC, Canada), with proteinase K permeabilization at 37 °C for 20 min. Slides treated with DNase1 were used as a positive control, while slides lacking the Terminal deoxynucleotidyl transferase enzyme solution during staining were used as the negative control. Slides were mounted as described in the immunofluorescence assay and imaged using confocal microscopy (Leica LSM880).

### 4.7. Western Blot Analysis

Protein lysates were prepared by mechanical shearing of frozen bladders by pestle and mortar, followed by protein extraction for 30 min at 4 °C in cold RIPA buffer. Lysates were centrifuged, and supernatant was collected and stored at −80 °C. Samples were diluted to 30 ng/L and heated at 95 °C for 5 min in 4× loading buffer. Western blots were performed on 7% (for collagen I and III) or 10% (for alpha-smooth muscle actin) SDS gels run under reducing conditions for 1 h. Protein was then transferred to PVDF membranes at 100 V for 1.5 h. Membranes were washed in PBS and blocked in 3% BSA before incubating with primary antibodies for goat anti-Collagen I (Southern Biotech), goat anti- Collagen III (Southern Biotech, Birmingham, AL, USA) 1/1000, a-SMA (Abcam) 1/5000, and GAPDH (Invitrogen, Waltham, MA, USA) 1/10,000 overnight at 4 °C. Membranes were then washed and incubated with an HRP-conjugated secondary antibody (Santa Cruz, Dallas, TX, USA) 1/5000 for 1 h. Membranes were washed and exposed to peroxide substrate for 5 min before imaging using the Amersham Imager 600. Quantification was done using Fiji.

### 4.8. Cystometry

Cystometry experiments were performed as previously described and modified for mice [32]. A catheter (PE10) was surgically inserted into the dome of the mouse bladder and channeled subcutaneously to the base of the neck. After 2 days of recovery, cystometry experiments were performed. Each catheter was connected to a transducer that recorded bladder pressure. A saline pump infused saline solution into the bladder at a rate of 1.5 mL/h. Volume of urine and frequency of micturition were performed continuously over 1 h. Maximal bladder pressure, threshold pressure, basal pressure, inter-micturition pressure, inter-contraction interval and urine volume were measured. Bladder capacity, micturition volume, and residual volume were calculated from the aforementioned measurements.

### 4.9. Statistical Analysis

Western blot analysis, proliferation, and immunofluorescent studies were analyzed using an unpaired *t*-test with 2 tailed distributions. For cystometry, the Mann-Whitney U test was used to analyze effects between genotypes. Statistical significance was defined as *p* < 0.05.

## Figures and Tables

**Figure 1 ijms-22-12387-f001:**
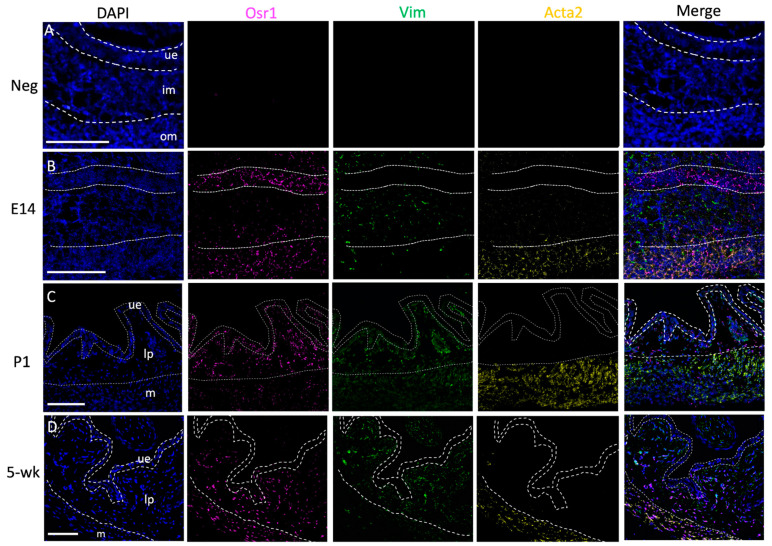
Osr1 shows dynamic changes in expression from the embryonic to adult bladder (**A**) Negative control using a scrambled probe (**B**) At E14 Osr1 is strongly expressed in epithelial and muscle cells and a few cells the lamina propria. (**C**) At P1 Osr1 is expressed in some cells in the epithelia, most cells in the lamina propria and muscle layer. (**D**) By five weeks of age, Osr1 is seen in most cells of the lamina propria and the majority of cells in the muscle layer. Rare epithelial cells showed expression. Odd-skipped related 1, *osr1* mRNA (pink), vimentin, *vim* (green), alpha smooth muscle actin, *acta2* (yellow), DAPI (blue). Scale bar = 100 μm *n* = 3 embryo/mouse for each time-point. Dotted lines are used to demarcate the three layers of the bladder: ue = uroepithelium, im = inner mesenchyme, om = outer mesenchyme, lp = lamina propria, m = muscle.

**Figure 2 ijms-22-12387-f002:**
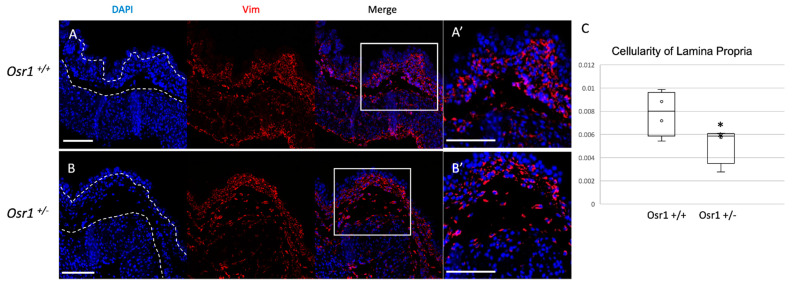
*Osr1^+/−^* newborn mice have a decreased number of cells in the lamina propria: Immunofluorescent staining for Vimentin protein (red) and DAPI (blue) shows more Vimentin-positive cells in the lamina propria of the Osr1^+/+^ mouse bladder (**A**,**A’**) compared to Osr1^+/−^ (**B**,**B’**). White boxes indicate magnified region in A’ and B’. (**C**) Cellularity was quantified by counting Vimentin-positive cells normalized to total lamina propria area in each section, asterisk (*) indicates *p* < 0.05. Scale bar = 100 μm (*n* = 4 mice/genotype, two to three sections per mouse were examined). Dotted lines delineate lamina propria.

**Figure 3 ijms-22-12387-f003:**
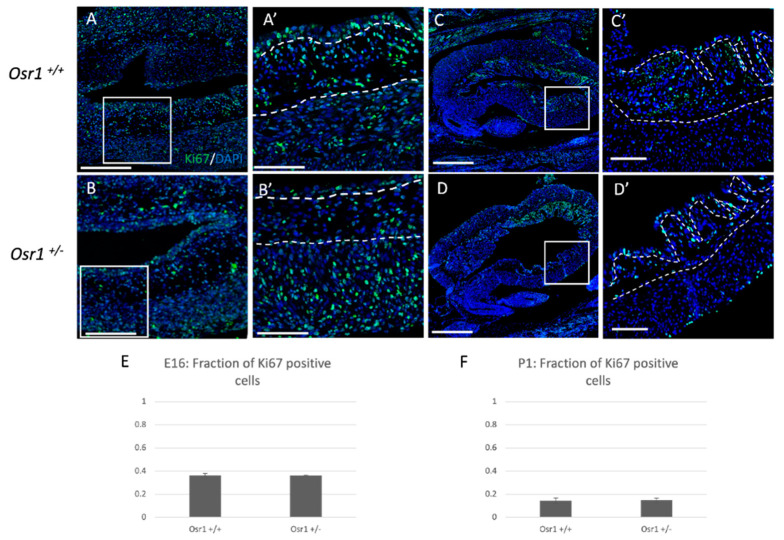
*Osr1^+/+^* and Osr1^+/−^ mouse bladders at E16 andP1 have similar levels of cell proliferation: Immunofluorescent staining for Ki67 protein (green) and DAPI (blue) in Osr1^+/+^ and Osr1^+/−^ mouse bladders at E16 (**A**,**A’**,**B**,**B’**) and P1 (**C**,**C’**,**D**,**D’**) are shown. White boxes identified magnified regions for A’–D’. (**E**,**F**) Graphs depict the fraction of Ki67-positive cells divided by number of DAPI-positive cells at E16 and P1, respectively. Scale bar A,B = 250 μm; Scale bar A’–D’ = 100 μm, and Scale bar C,D = 500 μm. White dotted lines delineate the borders between the epithelial, lamina propria, and muscle layers (*n* = 3 bladders examined/genotype, three sections per bladder were quantified in a fixed area, no statistically significant differences were seen.

**Figure 4 ijms-22-12387-f004:**
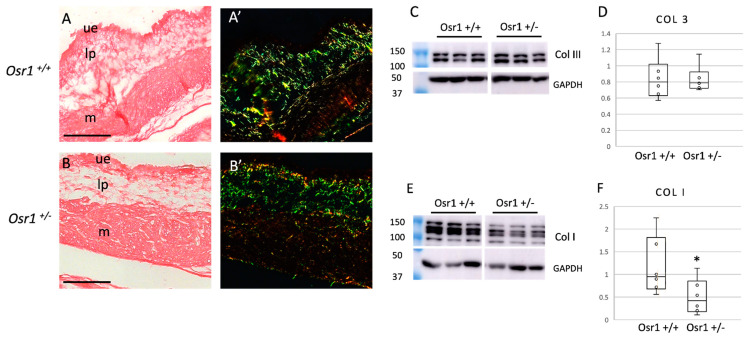
*Osr1^+/−^* newborn mouse bladders have decreased collagen I: Sirius red staining of Osr1^+/+^ (**A**,**A’**) and Osr1^+/−^ (**B**,**B’**) mice at the newborn stage show a decrease in collagen in Osr1^+/−^ mice. Under a polarized light filter (**A’**,**B’**) a decrease in thick (yellow) fibers is seen in the bladders of Osr1^+/−^ mice. Western blots of whole bladder lysates (**C**,**E**) show a decrease in total levels of Collagen I but not Collagen III in *Osr1^+/−^* mice. Quantitative analysis (**D**,**F**) shows collagen normalized to GAPDH protein levels. There is a statistically significant decrease in Col I protein expression in *Osr1^+/−^* mice, asterisk (*) indicates *p* = 0.05. *n* = 6 bladders/genotype. Scale bars = 100 μm. ue = uroepithelial layer, lp = lamina propria, m = muscle.

**Figure 5 ijms-22-12387-f005:**
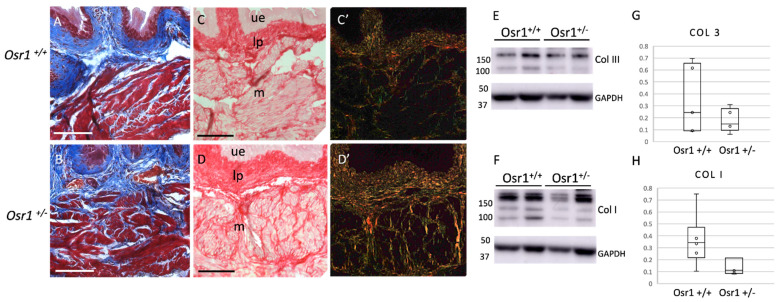
*Osr1^+/−^* adult mouse bladders have an increase in collagen: Masson’s trichrome staining of bladder sections of *Osr1^+/+^* (**A**) and *Osr1^+/−^* (**B**) mice. There is an increase in collagen (blue) in the muscle layer of Osr1^+/−^ mice. Sirius red of *Osr1^+/+^* (**C**,**C’**) and Osr1^+/−^ (**D**,**D’**) also show increased collagen between muscle bundles in Osr1^+/−^ mice, *n* = 3 bladders/genotype. Quantitative analysis of western blot showing Col III (**E**,**G**), and Col I (**F**,**H**) protein normalized to GAPDH. There are comparable levels of collagens in both Osr1^+/+^ and Osr1^+/−.^ Scale bars = 100 μm. ue = uroepithelial layer, lp = lamina propria, m = muscle.

**Figure 6 ijms-22-12387-f006:**
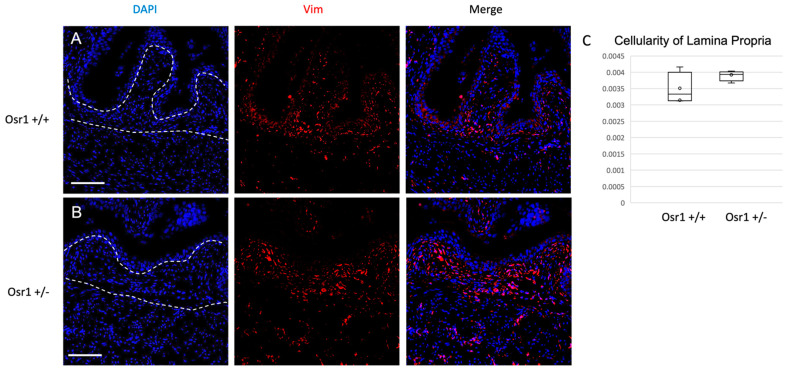
*Osr1^+/−^* and Osr1^+/−^ adult mice have similar number of cells in the lamina propria: Immunofluorescent staining for Vimentin protein (red) and DAPI (blue) shows a similar number of Vimentin-positive cells in the lamina propria of the Osr1^+/+^ mouse bladder (**A**) compared to Osr1^+/−^ (**B**). (**C**) Cellularity was quantified by counting Vimentin-positive cells normalized to total lamina propria area in each section (*n* = 3 mice/genotype). Scale bar = 100 μm. Dotted lines delineate lamina propria.

**Table 1 ijms-22-12387-t001:** Comparison between cystometry parameters in the Osr1+/+ and Osr1 +/− mice.

	Osr1^+/+^	Osr1^+/−^	*p* Value
Basal pressure (cm H_2_O)	11.42 (6.50)	13.58 (9.09)	0.758
Intermicturition pressure (cm H_2_O)	26.06 (8.66)	23.78 (13.97)	0.46
Threshold pressure (cm H_2_O)	44.94 (12.20)	44.17 (20.44)	0.951
Maximum pressure (cm H_2_O)	110.66 (32.84)	97.08 (51.85)	0.389
Micturition volume (mL)	**0.065 (0.033)**	**0.032 (0.007)**	**0.001 ***
Intercontraction interval (seconds)	**165.06 (54.57)**	**82.34 (26.37)**	**<0.001 ***
Spontaneous activity (cm H_2_O)	14.64 (5.03)	10.20 (9.39)	0.065
Bladder capacity (mL)	**0.075 (0.025)**	**0.036 (0.011)**	**<0.001 ***
Residual volume (mL)	0.013 (0.007)	0.007 (0.008)	0.121
Bladder compliance (mL/cm H_2_O)	**0.003 (0.001)**	**0.002 (0.001)**	**0.018 ***

Data are presented as mean (+/−SD) * *p* value < 0.05 using Mann-Whitney test.

## Data Availability

The data presented in this study are available in: Murugapoopathy V.; Cammisotto P.G.; Mossa A.; Campeau L.; Gupta I. R. Osr1 is Required for Mesenchymal Derivatives that Produce Collagen in the Bladder. *Int. J. Mol. Sci.*
**2021**, *22*, 12387. https://doi.org/10.3390/ijms222212387, as well as the corresponding Appendix A.

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
