# Peer review of "Osr1 Is Required for Mesenchymal Derivatives That Produce Collagen in the Bladder"

_ijms, 2021, doi:10.3390/ijms222212387_

Round 1
Reviewer 1 Report
This is a very interesting article that looks at the link between the transcription factor OSR-1 and fibrosis. The authors claim that OSR-1 induces collagen accumulation which in turn induces bladder dysfunction following injury.
This article is clear and well written. The results were well within the authors' hypotheses. This article is in line with the IJMS publication. It is important and should be published.
Major revisions
The points to review are;
It is surprising to have a discussion that is hardly based on the literature. The discussion needs to be completely rewritten and justify how these results are consistent with the literature. Also each hypothesis should be supported by previous publications.
Two points need to be developed and should be supported by results if possible. It is necessary to demonstrate or justify better by which mechanisms the newborns have a decrease of collagens and on the contrary the adults have an accumulation.
Similarly, the mechanisms linking OSR1 to bladder dysfunction should be supported by results if possible, if not by hypotheses on the mechanisms based on the previous work of other teams.
Reviewer 2 Report
The manuscript by Murugapoopathy V et al addresses the role of the Osr1 transcription factor in collagen production/deposition in the bladder. By using mice bearing the heterozygous deletion of Osr1, the authors show that Osr1 regulates fibroblast number in the lamina propria of bladder wall, and collagen production in the bladder from embryonic development to adulthood. The study is well performed and data are clearly explained throughout the manuscript. The data provided are of interest for the readers of the International Journal of Molecular Sciences. Despite this, further experiments are necessary to support the authors’ conclusions.
General comment: the authors should avoid to overconclude. Only data-driven conclusions should be included in the text. Speculations are ok if they are clearly presented as speculations or hypothesis. Please go through the text and correct accordingly (starting from the title) (examples of over-conclusions are: lines 210-212; lines 226-227). In this context, please clearly state that Figure 6 is a working hypothesis rather than a summary of findings.
Major points
- The authors should demonstrate more thoroughly that Osr1+/- mice have a decreased number of fibroblasts/mesenchymal progenitors in the bladder wall. In particular, the authors should characterize the cell population that is decreased in Osr1+/- mice by further IF or in situ hybridization (ISH) experiments. The only detection of vimentin is not sufficient. Please use further markers typical of mesenchymal progenitors to demonstrate that Osr1 is involved in the regulation of mesenchymal progenitors.
- The authors should follow the evolution during development and growth (E14, P1, 5wk) of collagen expression and deposition by simultaneously monitoring the expression at mRNA level and the accumulation at protein level of ColI and ColIII by ISH and IF/Sirius red respectively. Co-staining with vimentin or other fibroblast markers would allow to correlate the changes in collagen amount to the reduction of fibroblasts and would clarify the PCR results shown in suppl Figure 3. Also, these experiments would provide a clearer explanation of the “paradoxical” increase in collagen that the authors found in adult Osr1+/- mice.
Minor points:
Figure 2: the authors should normalize the number of vim+ cells to the total number of cells (nuclei count) in the lamina propria, rather than to the total area of lamina propria.
Figure 5: (same as major point 2) could the authors assess vimentin expression in adult bladder samples? I think this could help to understand whether the reduced number of fibroblasts is maintained through adulthood and would provide insights about the mechanisms underlying the increase in collagen in adult Osr1+/- mice.
Figure 5E-F: please test additional samples to confirm that the levels of ColI-III are unchanged in Osr1+/- and control mice.
Figure Suppl 1: From the images provided a-SMA expression seems increased in Osr1+/- mice. Please provide image/WB quantification to confirm the authors’ statement or correct the text accordingly.
Round 2
Reviewer 1 Report
Accepted for publication
Reviewer 2 Report
The authors have addressed all my previous comments.